# Targeted Therapies in Advanced and Metastatic Urothelial Carcinoma

**DOI:** 10.3390/cancers14215431

**Published:** 2022-11-04

**Authors:** Andrew B. Katims, Peter A. Reisz, Lucas Nogueira, Hong Truong, Andrew T. Lenis, Eugene J. Pietzak, Kwanghee Kim, Jonathan A. Coleman

**Affiliations:** 1Urology Service, Department of Surgery, Memorial Sloan Kettering Cancer Center, New York, NY 10065, USA; 2Department of Surgery, Memorial Sloan Kettering Cancer Center, New York, NY 10065, USA

**Keywords:** metastatic urothelial carcinoma, molecular targeted therapy, tumor biomarkers, antineoplastic therapy, monoclonal antibodies, antibody-drug conjugates, immunotherapy, next-generation sequencing

## Abstract

**Simple Summary:**

Urothelial carcinoma is a malignancy of the cells lining the genitourinary tract but it most commonly occurs in the bladder. Once urothelial carcinoma has spread outside of the genitourinary tract, survival outcomes are poor despite the standard of care treatment of chemotherapy followed by immunotherapy. Genetic sequencing of cancer tissue has identified targets for new anti-cancer drugs. This review summarizes the evidence regarding the efficacy of targeted therapies in advanced urothelial carcinoma.

**Abstract:**

This review describes the current landscape of targeted therapies in urothelial carcinoma. The standard of care for advanced urothelial carcinoma patients remains platinum-based combination chemotherapy followed by immunotherapy. However, median overall survival for these patients is still <1 year and there is an urgent need for alternative therapies. The advent of next-generation sequencing has allowed widespread comprehensive molecular characterization of urothelial tumors and, subsequently, the development of therapies targeting specific molecular pathways implicated in carcinogenesis such as FGFR inhibition, Nectin-4, Trop-2, and HER2 targeting. As these therapies are demonstrated to be effective in the second-line setting, they will be advanced in the treatment paradigm to localized and even non-muscle invasive disease.

## 1. Introduction

Advanced and metastatic urothelial carcinoma (mUC) of the bladder comprises a small subset of all urothelial tumors but accounts for the majority of the rapid mortality associated with this disease. In the mUC setting, prior to combination cisplatin-based chemotherapy, median survival was only approximately 4–6 months [1]. The development of cisplatin-based combination chemotherapy regimens increased median survival to 11–14.8 months with objective response rates (ORR) in the range of 40–49% [2,3,4]. While an improvement, relapse and progression are still very common in this disease and second line chemotherapies have demonstrated poor overall activity [5]. In addition, a significant portion (30–50%) of this patient population is cisplatin-ineligible due to renal insufficiency, poor performance status, hearing loss, neuropathy, or heart failure emphasizing the need for alternative, more efficacious therapies [6].

Over the last decade, accelerating basic science research has rendered a deeper understanding of the molecular biology of urothelial tumors, leading to the development of novel treatment strategies. Urothelial tumors were found to have a high mutational burden and to express high levels of programmed cell death ligand 1 (PD-L1), which when bound to its PD-1 receptor results in suppression of the T-cell-mediated antitumor immune response allowing immune escape and tumor progression [7]. High PD-L1 expression has been strongly associated with a poor prognosis in urothelial carcinoma [8]. Antibodies designed to block this interaction and other immune checkpoint inhibitors (CPI) have demonstrated encouraging results in the second-line treatment of mUC as well as in treatment-naïve cisplatin-ineligible patients with some durable responses and a favorable toxicity profile when compared to further chemotherapy. Phase III trials (NCT02302807, NCT00256436) of atezolizumab (Tecentriq^®®^) and pembrolizumab (Keytruda^®®^) in the second-line setting with an ORR of 20.3–23% and median OS 10.1–11.1 months led to Food and Drug Administration (FDA) approval of these immune checkpoint inhibitors starting in 2016 [9,10]. This was followed by approval of nivolumab (Opdivo^®®^), durvalumab (Imfinzi^®®^), and avelumab (Bavencio^®®^) based on Phase I/II study data (NCT02387996, NCT01693562, NCT01772004) [11,12,13]. However, CPI as second-line therapy is still associated with a median overall survival <1 year. Consequently, multiple trials have been initiated evaluating combination regimens with dual CPI, CPI plus chemotherapy, or CPI plus alternative novel agents targeting specific molecular pathways [14,15]. This review will focus on these targeted therapies in advanced or metastatic urothelial cancer. We will discuss the molecular characterization of urothelial carcinoma that has led to identification of therapeutic targets. We will then review data from studies evaluating novel antibody and antibody-drug conjugates targeting pathways known to be critical in tumor development. Finally, we will discuss the limitations of tumor sequencing and targeted therapy and highlight opportunities for future research.

## 2. Next-Generation Sequencing Identifies Therapeutic Targets

Characterization of clinically relevant molecular pathways in urothelial carcinoma has been ongoing for many years but has exponentially accelerated with the advent of next-generation sequencing (NGS) and the completion of The Cancer Genome Atlas (TCGA) project providing a comprehensive profile of 131 high-grade muscle-invasive urothelial tumors of the bladder in 2014 [16]. Using whole exome sequencing (WES) with matched normal samples, 32 genes demonstrated significant levels of recurrent somatic mutation with TP53 pathway inactivation in 76% of the samples. Alterations affecting the PI3K/Akt/mTOR were found in 42% of samples and MAPK pathway alterations in 44% of samples [16]. In 2017, the TCGA published updated results from 412 muscle-invasive bladder cancers, identifying 58 significantly mutated genes (Figure 1) [17]. Subsequent sequencing studies have classified urothelial carcinoma into distinct molecular subtypes with differential prognoses and responses to conventional therapy [18]. Based on these large-scale molecular studies, targeted therapies have been developed and are just now demonstrating their efficacy in clinical trials (Table 1).

Several transcriptomic studies have further characterized the heterogeneity of mUC and have allowed for more accurate molecular characterization which has improved prognostic stratification and improved targeted therapeutic options. In 2020, Kamoun et al. reconciled 18 datasets to derive a consensus molecular classification consisting of 6 distinct molecular subtypes which were labelled as luminal papillary (LumP), luminal nonspecified (LumNS), luminal unstable (LumU), stroma-rich, basal/squamous (Ba/Sq), and neuroendocrine-like (NE-like). Each subtype has a distinct clinical, histologic, mutational, survival, and stromal/immune infiltrate as summarized in Figure 2 [18]. While these subtypes are critical to characterizing tumors and targeting therapies, it is important to note that the differences in intertumoral heterogeneity do not necessarily reflect differences in intratumor heterogeneity and there are many overlapping signaling pathways that can be targeted (Figure 3). In each subsequent section, we will highlight the molecular classification associated with each targeted therapy.

## 3. Fibroblast Growth Factor Receptor (FGFR) Inhibition

The FGFR signaling pathway is involved in cell proliferation, differentiation, angiogenesis, metabolism, mobility, and invasion. Five FGF receptors have been identified (FGFR1-5). Cell signaling is propagated via a tyrosine kinase signaling pathway. Activation of FGFR causes four main downstream cascades which include RAS/Raf/MEK-MAPK, PI3K/AKT, PLC-gamma, and signal transducer and activation of transcription (STAT) [33,34]. *FGFR* alterations have been identified most commonly in urothelial carcinoma and intrahepatic cholangiocarcinoma.

FGFR3 alterations are particularly common in urothelial cancer and have been noted in 12–18% of urothelial tumors and are generally associated with lower grade and lower stage tumors. However, up to 20% of advanced urothelial carcinoma and 37% of upper tract urothelial carcinoma (UTUC) demonstrate FGFR3 alterations. FGFR3 alterations have been associated with the LumP subtype and a T-cell-depleted immune phenotype in bladder cancer which may contribute to a poor response to CPI [35,36]. Of note, the LumP subtype is associated with the best median overall survival (4 years) and a papillary architecture. Both FGFR3-specific and pan-FGFR inhibitors have been developed with variable activity. Erdafitinib (Balversa^®®^), a tyrosine kinase (TKI) FGFR1-4 inhibitor, demonstrated anti-tumor activity in preclinical models as well as a Phase I trial (NCT01703481) in patients with known somatic FGFR3 alterations for patients with mUC or intrahepatic cholangiocarcinoma. There was a 46% response rate in UC and a 27% response rate in patients with cholangiocarcinoma [37]. This led to a Phase II trial (NCT02365597) and FDA approval of the use of erdafitinib for FGFR2/3-altered platinum-treated mUC patients with an ORR of 40% and median OS of 13.8 months. The most common side effects included: hyperphosphatemia (77%), stomatitis (58%), and diarrhea (51%). Among the 67% of grade 3 or 4 events, 46% were considered to be treatment related and consisted of hyponatremia (11%), stomatitis (10%), and asthenia (7%) [19]. Erdafitinib also has FDA approval in the setting of locally advanced, unresectable, or metastatic cholangiocarcinoma and known FGFR2 mutations. Infigratinib (Truseltiq^®®^), a selective inhibitor of FGFR1-3, was originally developed to treat FGFR2-mutated advanced cholangiocarcinoma but has demonstrated activity in small prospective trials in previously treated FGFR-altered mUC with an ORR of 25.3% and a notably higher ORR of 50% in a subset of eight patients with UTUC [20,38]. A Phase III trial (NCT04197986) is planned for mUC, and a Phase I trial is recruiting for treatment of high-grade glioma (NCT04424966). Rogaratinib is a potent small molecule inhibitor of FGFR1-4 kinase activity whose activity has been strongly correlated to high tumor FGFR mRNA expression, with mRNA expression used as an alternative and potentially more specific biomarker in a Phase I study (NCT01976741). This study included 52 patients with urothelial carcinoma with an ORR of 23%. Other solid tumors with high levels of FGFR expression were included in this study, such as non-small cell lung cancer (NSCLC), head and neck squamous cell carcinoma (HNSCC), and cholangiocarcinoma, but none had as robust of a response compared to urothelial carcinoma [21]. An ongoing Phase II/III trial (NCT03410693) recently published unplanned interim analysis results which compared Rogaratinib to chemotherapy (docetaxel, paclitaxel, or vinflunine). The ORR was 20.7% for the Rogaratinib group vs. 19.3% for the chemotherapy group. Because of the similar efficacy, the trial was stopped prior to progression to Phase III [22]. While better tolerated than cytotoxic chemotherapy, FGFR3 inhibitors are not without adverse events, the most common being hyperphosphatemia, diarrhea, fatigue, central serous retinopathy, stomatitis, and nail/skin disorders [39]. FGFR3 inhibition serves as a pivotal and first step in the approval of targeted therapies in urothelial cancers based on the unique molecular alterations present in an individual tumor. It is noteworthy that the activation signatures associated with the LumP subtype suggest that these tumors may respond to FGFR inhibition regardless of their FGFR3 mutational status [17,40].

## 4. Nectin-4 Targeting

Nectin-4 is an immunoglobulin-like adhesion molecule that acts as a tumor antigen that is overexpressed in most epithelial cancers, including urothelial carcinomas [41]. Nectin-4 has been shown to promote cancer cell proliferation and metastasis through activation of the PI3K-AKT signaling pathway. Additionally, there is interaction with ERBB2, a tyrosine kinase which activates PI3K-AKT, which has important therapeutic implications, which will be discussed in a later section [42]. Enfortumab vedotin (EV) (Padcev^®®^) is an antibody-drug conjugate (ADC) with a human anti-nectin-4 antibody linked to monomethyl auristatin E, which has cytotoxic activity via microtubule disruption. In 2019, EV was approved by the FDA based on results from a Phase I trial (NCT02091999) in mUC patients previously treated with chemotherapy that demonstrated an ORR of 43% and a median OS of 12.3 months. This efficacy was further supported by the recently published Phase III data (NCT03474107) from the EV-301 trial of EV versus chemotherapy demonstrating an ORR of 40.6% vs. 17.9% and a median OS 12.8 vs. 8.9 months, respectively [23]. Treatment-related adverse events (TRAE) were common, with 51.4% of the EV group experiencing a grade 3 or higher TRAE. The most common adverse events were maculopapular rash, fatigue, and neutropenia [23]. Notably, Nectin-4 expression is considered widespread enough that no biomarker testing is needed prior to treatment. However, recent analysis of Nectin-4 expression across molecular subtypes demonstrated heterogeneity with significant enrichment of Nectin-4 expression in luminal subtypes (LumP, LumNS, LumU) and they had a positive correlation with luminal subtype molecular drivers such as GATA3, PPARG, and FOXA1 [43]. More research is needed to assess if patients harboring non-luminal subtypes show equal benefit to EV compared to the luminal counterparts. EV is not approved for any other malignancy at this time.

Combination therapy of EV with pembrolizumab, a PD1 receptor inhibitor, in the Phase Ib/II EV-103 trial (NCT03288545) has demonstrated promising results with an ORR of 73.3% in a heavily pre-treated population with one third of patients having liver metastases [24]. Combination therapy with EV plus pembrolizumab is also being investigated in the peri-operative setting in EV-303, a Phase II study (NCT03924895) [44].

## 5. Trophoblast Cell Surface Antigen 2 (Trop-2) Targeting

Trop-2 is a transmembrane protein that has been shown to be overexpressed in urothelial carcinoma, acting as a tumor antigen. Activation of Trop-2 increases intracellular calcium stores, which activates MAPK signaling, as well as the NF-KB and RAF pathways [45]. Sacituzumab govitecan (SG) (Trodelvy^®®^) is an ADC consisting of an anti-Trop-2 antibody linked to SN-38 which is the active metabolite of irinotecan [45]. SG demonstrated efficacy in metastatic triple negative breast cancer, NSCLC, and urothelial cancer [46]. The TROPHY-U-01 Phase II registrational trial (NCT03547973) of SG in previously treated mUC demonstrated an ORR of 27% and a median OS of 10.5 months. The multicenter Phase III trial (NCT04527991) TROPiCS-04 comparing SG versus single-agent chemotherapy treatment of choice is currently enrolling [25]. The most significant toxicity associated with SG is myelosuppression, along with alopecia, and diarrhea [46]. Neoadjuvant SG with or without pembrolizumab prior to radical cystectomy is under investigation in the Phase II SURE trial (NCT05226117) [47]. Originally, SG received FDA approval for treatment of metastatic triple negative breast cancer. Subsequently, on 13 April 2021, the FDA granted accelerated approval for SG for patients with mUC previously treated with platinum-based chemotherapy or immunotherapy and it was also fast-tracked for treatment of NSCLC. Furthermore, a recent analysis found high Trop-2 expression in all molecular subtypes except neuroendocrine. Tumor cells retained expression of Trop-2 even after prolonged exposure to EV, suggesting feasibility of a second line treatment with SG after EV [48].

## 6. Human Epidermal Growth Factor 2 (HER2) Targeting

HER2 (encoded by gene *ERBB*2) is a non-ligand binding member of the human epidermal growth factor receptor family of receptor tyrosine kinase involved in the mitogen-activated protein kinase (MAPK) signaling pathway. NGS has demonstrated high rates of *ERBB* gene alterations and HER2 overexpression in urothelial carcinoma [17,49]. The ERBB family consists of EGFR, HER2 (ERBB2), ERBB3, and ERBB4. HER2 expression is most commonly seen in the LumU subtype and EGFR expression is seen more commonly in the Ba/Sq subtype. Initial small trials using HER2 directed therapies in mUC have demonstrated mixed results. Afatinib (Gilotrif^®®^, Giotrif^®®^, Afanix^®®^), an oral irreversible inhibitor of the ERBB family originally approved for EGFR-altered NSCLC patients, was studied in a Phase II trial (NCT02122172) in platinum-refractory mUC and demonstrated some efficacy in PFS at 3 months, with five of the six patients with HER2 and ERBB3 alterations meeting the endpoint [50]. Afatinib has also shown some activity against HNSCC and metastatic breast cancer but has not yet received FDA approval. Trastuzumab (Herceptin^®®^, Ogivri^®®^, Herzuma^®®^), an anti-HER2 monoclonal Ab, was originally developed for HER2+ metastatic breast cancer but has subsequently been approved for gastric adenocarcinoma and gastroesophageal junction adenocarcinoma. A Phase II study (NCT02091141) evaluated the efficacy of trastuzumab plus pertuzumab (Perjeta^®®^), an anti-HER2 monoclonal Ab with complementary binding kinetics used in HER2+ breast cancer, in previously treated mUC with an ORR of 33% in those with HER2 overexpression [27]. However, use of HER2-targeted ADCs, such as RC48-ADC, has demonstrated some promise, demonstrating an ORR of 51.2% in previously treated patients with mUC, with an ORR of 60% in a subgroup of patients with higher HER2 expression (NCT03507166) [28]. Toxicity was low, with leukopenia, hypoesthesia, and alopecia being most common [28]. Trastuzumab linked to deruxtecan (Enhertu^®®^), a topoisomerase I inhibitor, is another ADC with similar indications to trastuzumab monotherapy and is currently being tested in combination with nivolumab (NCT03523572). Preliminary results reveal an ORR of 36.7% with a median OS of 11 months. The safety profile was similar to those treated with trastuzumab deruxtecan alone, with interstitial lung disease/pneumonitis occurring in 23.5% of all participants [26]. SGNTUC-019 is an ongoing Phase II basket trial (NCT04579380) with dual targeting of HER2 with tucatinib (Tukysa^®®^), a HER2 kinase inhibitor also used in the treatment of HER2+ breast cancer, and trastuzumab in patients with previously treated solid tumors, including urothelial carcinoma [51].

## 7. Transforming Growth Factor Beta (TGF-β) Inhibition

TGF-β is a cytokine that can act to suppress the host immune response to tumor formation and contribute to anti-PD-L1 resistance in mUC by excluding CD8 T cells from the tumor [52]. There is also evidence that TGF-β plays a role in regulating the epithelial–mesenchymal transition (EMT) of cancer cells, which is associated with invasion and therapy resistance [53]. Preclinical studies of TGF-β inhibition demonstrated improved anti-tumor immunity; however single agent anti-TGF-β therapy alone has not been very successful. Bintrafusp alfa (BA) is a first-in-class bifunctional anti-PD-L1/TGF-β receptor II fusion protein designed to inhibit PD-L1-mediated immunosuppression as well as TGF-β levels in the tumor microenvironment. Preclinical data demonstrated that BA resulted in improved antitumor activity vs. TGF-β or PD-L1 monotherapy alone and was able to both prevent EMT as well as revert cells that had already undergone EMT in multiple solid tumor types [53]. Two Phase I trials (NCT02699515, NCT02517398) in solid tumors demonstrated an ORR of 14.3–26.5% with a tolerable toxicity profile [29,30]. A urothelial-specific Phase Ib trial (NCT04349280) is ongoing in previously platinum-treated mUC patients with the goal of enrolling 40 patients [54]. However, a Phase II trial (NCT04501094) in patients with metastatic, refractory urothelial carcinoma was terminated due to low accrual. Trials involving BA have lacked improved efficacy over established therapies which may halt future development.

## 8. Phosphoinositide-3-Kinase (PI3K) Inhibition

The PI3K pathway is upstream of the protein kinase B (Akt) and mammalian target of rapamycin (mTOR) pathways, with PIK3CA mutations and subsequent upregulation of the PI3K pathway common in urothelial carcinoma. The PI3K pathway is activated due to multiple different mutations as previously discussed, and because of its ubiquity, regulation of apoptosis, and multiple downstream targets, we will review the signaling pathway in greater depth.

PI3K is activated by extracellular signals (i.e., FGFR, EGFR, receptor tyrosine kinase, etc.) which leads to autophosphorylation and activation of Akt through PIP2 to PIP3 conversion. Activation of Akt leads to several events. Primarily, mTOR is upregulated which in turn increases cell growth, protein synthesis, and energy storage. There is a further direct effect on NF-KB. Phosphorylation of Xiap leads to inhibition of MDM2 and subsequent activation of p53 which blocks autophagy. There is also activation of PAK1 which has inhibitory effects on apoptosis. Ultimately, these events lead to cancer cell resistance (anti-apoptosis), cell cycle progression, and cell proliferation. The primary inhibitor of this pathway is the tumor suppressor, PTEN [55]. Further evidence suggests that PI3K serves as a gatekeeper, preventing excessive innate immune response [56].

It is thought that PIK3CA mutations are an early genetic alteration in urothelial carcinogenesis and are associated with FGFR3 mutations [57]. Eganelisib is a novel oral inhibitor of the PI3K-gamma subunit evaluated in a Phase II study (NCT03980041) in combination with nivolumab with an ORR of 30% in platinum-treated mUC. Because of the effect on the immune response, eganelisib is designed to increase the efficacy of nivolumab. Notably PD-L1+ patients demonstrated an ORR of 80%, although this was in a small subset of five patients [31]. Eganelisib is currently FDA approved for the treatment of triple negative breast cancer.

## 9. Mammalian Target of Rapamycin (mTOR) Inhibition

The mTOR pathway is downstream from the PI3K and Akt pathways. The mTOR pathway consists of mTORC1 or mTORC2 which are kinases involved in cell cycle regulation as well as angiogenesis via hypoxia-inducible factor-1 (HIF-1) and they act via the tuberous sclerosis complex 1 and 2 (TSC1/TSC2) GPTase activating protein (GAP). Everolimus (Zortress^®®^, Afinitor^®®^, Disperz^®®^), an oral mTORC1 inhibitor currently used to treat advanced renal cell carcinoma, HR+ HER2− breast cancer, neuroendocrine tumors, and tumors associated with tuberous sclerosis, was evaluated in a Phase II trial (NCT00805129) in platinum-treated mUC with an ORR of only 5%, although patients with TSC1-mutant tumors demonstrated a significantly better response [58,59]. Based on this finding, several additional Phase II trials (NCT01827943) using mTOR inhibitor monotherapy were initiated but have not yielded encouraging results to date [60]. Buparlisib, originally developed for HER2- breast cancer but abandoned due to high toxicity and suboptimal results, is a pan-isoform class I PI3K inhibitor evaluated in a Phase II study (NCT01551030) of 13 patients but it did not meet its primary endpoints even in TSC1-altered patients and was associated with significant toxicity and currently has no FDA approvals [61]. Sapanisertib is a potent inhibitor of mTORC1 and 2 and was recently evaluated in a Phase II trial (NCT03047213) in 17 patients with TSC1/TSC2-mutated mUC, but the trial was terminated for futility, and it was suggested that further studies should evaluate molecular alterations beyond TSC1/2 [32]. Sapanisertib is currently under investigation in multiple solid tumor types with no current FDA approvals. Combination therapy with TKIs, CPI, and mTOR inhibitors have been more promising, with multiple trials ongoing [62]. Treatment with TKIs is not without adverse events and many trials must limit dosing or end accrual early due to intolerable side effects such as edema, hypothyroidism, nausea, vomiting and diarrhea [63].

## 10. Limitations of Targeted Therapy

Despite the excitement surrounding targeted therapies, there remain significant limitations to their success. The first is intratumoral and temporal heterogeneity. Urothelial tumors are generally genetically unstable and heterogeneous in their clonal makeup [7]. A therapy that targets an alteration in one clone may be ineffective against others or even accelerate their growth by applying selective pressure. Similarly, metastatic or recurrent disease may have different somatic alterations than the primary with differential treatment response. Biomarkers derived from bulk sequencing or immunohistochemistry (IHC) of one biopsy may not be representative of the remainder of the disease burden. Combination therapy with cytotoxic agents, CPI, and targeted therapies seeks to combat treatment resistance driven by the selective pressure of monotherapies.

Another significant limitation is the cost and availability of NGS. Despite an exponential decrease in the cost and speed of sequencing, routine sequencing of cancer tissue has not yet gained traction in clinical practice [64]. Cancer care delivery is highly fragmented, and resources vary depending on geography. The institutional investment required to establish a clinical genomics program is high, including equipment, reagents, personnel, and an analytics team. There are commercial assays available, but they vary in expense, coverage, and targets identified. Protocols for specimen handling, storage, and transportation can be complex and labor-intensive. We expect that in the future, routine NGS will be commonplace, but there remain many systemic barriers.

On the horizon, there is an increasing role for single cell and spatial transcriptomics/proteomics along with computational methods to more comprehensively characterize tumors and their immune microenvironment to identify reliable and predictive biomarkers. These new technologies, while powerful, are currently cost prohibitive for use outside of the research setting and limited to highly specialized centers.

## 11. Conclusions

The advent of NGS has accelerated our ability to accurately characterize the molecular biology of urothelial carcinoma, allowing identification of distinct molecular subgroup classification, carcinogenic targets, and rapid iteration of potential therapeutic compounds. The standard of care remains platinum-based chemotherapy followed by maintenance immunotherapy or immunotherapy alone in platinum-ineligible patients. However, the relatively dismal prognosis of mUC and large proportion of platinum-ineligible patients emphasize the urgent and unmet need for additional therapies. The therapies reviewed above are promising both in the second line setting and in combination with CPI or chemotherapy. As these therapies are demonstrated to be efficacious in the metastatic setting, new trials have begun advancing these targeted therapies earlier in the treatment paradigm to the localized, neoadjuvant, and even non-muscle invasive setting.

Molecular characterization based on transcriptomic profiling has allowed for the identification of six distinct, clinically relevant consensus classes. Though there are unique transcriptomic signatures of each class, there is significant overlap in cell signaling pathways which may be targeted at various steps. Alterations leading to activation of the PI3K/AKT pathway are common among all luminal types and the Ba/Sq subtype. Yet, there is more to be explored from consensus classification. For example, NE-like and LumU tumors have expression profiles suggesting sensitivity to radiotherapy, and Ba/Sq appears to be sensitive to EGFR targeted therapies in pre-clinical models. These avenues are yet to be studied robustly in a clinical trial. Further complicating matters are intratumor heterogeneity and the tumor immune microenvironment regulating responses to specific targeted therapies.

Taken together, there is optimism that adoption of widespread molecular profiling, targeted therapies alone or in combination, and increased clinical trial availability and enrollment will result in meaningful improvements in survival for patients with urothelial carcinoma.

## Figures and Tables

**Figure 1 cancers-14-05431-f001:**
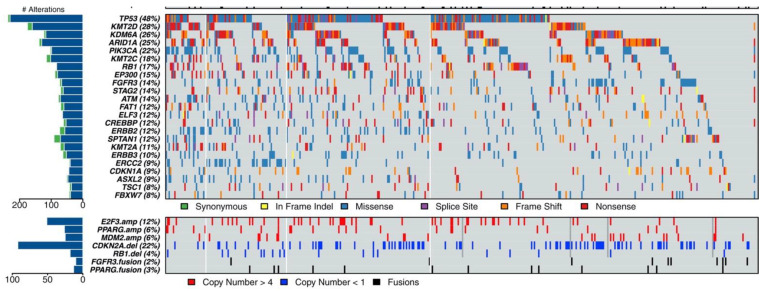
Mutational landscape of muscle-invasive urothelial carcinoma [17].

**Figure 2 cancers-14-05431-f002:**
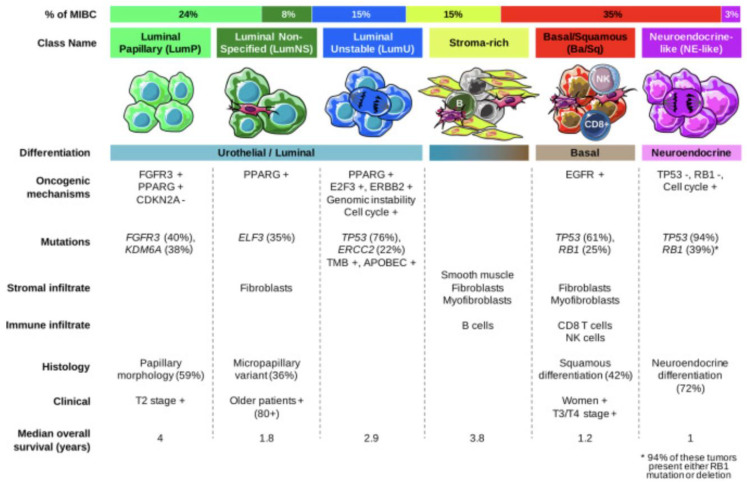
Summary of consensus classes and their unique features [18].

**Figure 3 cancers-14-05431-f003:**
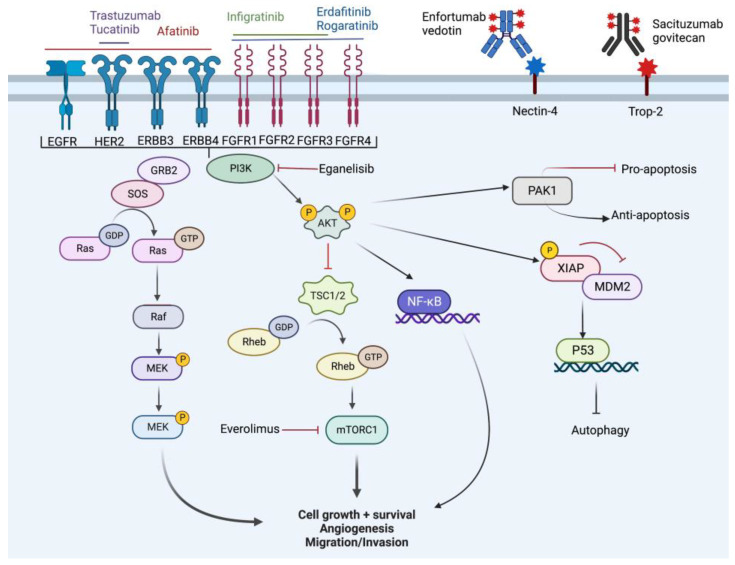
Overview of cell signaling pathway and targets of novel therapy. Created with BioRender.com (accessed on 18 October 2022).

**Table 1 cancers-14-05431-t001:** Clinical data supporting the efficacy of targeted therapies in urothelial carcinoma.

Reference	Agent	Trial	Phase	Setting	N	Biomarker	ORR	Median OS (months)	Primary Endpoint
Powles et al. Lancet 2018 [9]	atezolizumab vs. chemo	NCT02302807	3	2L mUC	931	PD-L1 (IC2/3)	23.0%	11.1 vs. 10.6	OS
Bellmunt et al. NJEM 2017 [10]	pembrolizumab vs. chemo	NCT02256436	3	2L mUC	542	PD-L1	21.1%	10.3 vs. 7.4	OS, PFS
Loriot et al. NEJM 2019 [19]	erdafitinib	NCT02365597	2	2L mUC	99	FGFR3	40.0%	13.8	ORR
Pal et al. Cancer Discov 2018 [20]	infigratinib (BGJ398)	NCT01004224	1	2L mUC	67	FGFR3	25.4%	7.75	ORR
Schuler et al. Lancet Oncol 2019 [21]	rogartinib	NCT01976741	1	2L mUC	52	FGFR3	23.0%	NR	Safety
Sternberg et al. JCO 2022 [22]	rogartinib vs. chemo	NCT03410693	2/3	2L mUC	175	FGFR3	20.7%	8.3	ORR, OS
Powles et al. NEJM 2021 [23]	enfortumab vedotin vs. chemo	NCT03474107	3	2L mUC	608	Nectin-4	40.6%	12.8 vs. 8.9	OS
Rosenberg et al. [abstract] JCO 2020 [24]	enfortumab vedotin + pembro	NCT03288545	1b/2	1L cis-ineligible	45	Nectin-4/PD-L1	73.3%	NR	Safety, ORR
Tagawa et al. JCO 2021 [25]	sacituzumab govitecan	NCT03547973	2	2L mUC	113	Trop-2	27.0%	10.5	ORR
Galsky et al. [abstract] JCO 2022 [26]	trastuzumab + nivolumab	NCT03523572	1b/2	2L mUC	34	HER2/PD-L1	36.7%	11	ORR
Bryce et al. [abstract] JCO 2017 [27]	trastuzumab + pertuzumab	NCT02091141	2	2L mUC	12	HER2	33.0%	NR	ORR
Sheng et al. Clin Cancer Res 2021 [28]	anti-HER2 RC48-ADC	NCT03507166	2	2L mUC	43	HER2	51.2%	13.9	ORR
Doi et al. Oncologist 2020 [29]	bintrafusp alfa	NCT02699515	1	2L solid tumors	23	TGF-Beta/PD-L1	14.3%	NR	Safety
Strauss et al. Clin Cancer Res 2018 [30]	bintrafusp alfa	NCT02517398	1	2L solid tumors	19	TGF-Beta/PD-L1	26.5%	NR	Safety
Tomczak et al. [abstract] JCO 2021 [31]	eganelisib + nivolumab	NCT03980041	2	2L mUC	49	PI3K-y	30.3%	NR	ORR
Kim et al. [abstract] JCO 2021 [32]	sapanisertib	NCT03047213	2	2L mUC	17	mTORC1/2	0.0%	3.4	ORR

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
