# Peer review of "Targeted Therapies in Advanced and Metastatic Urothelial Carcinoma"

_cancers, 2022, doi:10.3390/cancers14215431_

Round 1

Reviewer 1 Report (Previous Reviewer 2)

Journal MDPI Cancers -Open Journal since 2009 IF = 6.129 (2019)

Title: “Targeted therapies in urothelial carcinoma – Review Article”

Authors: Andrew B. Katims, Peter A. Reisz, Lucas Nogueira, Hong Truong, Andrew T. Lenis, Eugene J. Pietzak, Kwanghee Kim, and Jonathan A. Coleman

Affiliation: Memorial Sloan Kettering Urological Services

Overall Impression: The manuscript has been improved with revisions based on the previous reviewers’ comments and concerns. The presentation of the new agents alone or in combination, and their rationale for use, and ongoing clinical trial investigations is helpful to readers with an interest in urothelial cancer.

Major Concerns: It would be nice to know how new or novel drugs have been applied to other cancers and their response to help assess if there is a specific advantage to urothelial cancer with therapy being directed by next gen sequencing (NSG). This has been brought up before by the reviewers wanting to know if the drug was developed specifically for UC or another cancer. Based on the title “targeted therapies for urothelial cancers” this not beyond the scope of this paper as a significant proportion of drugs were developed for other indications.

Minor Concerns: Being directed at urologists when the article should be directed at the multidisciplinary team caring for patients with mUC.

Line by line Review:

Page 1, line 25: “Urologists must be prepared to be involved in the multi-disciplinary care of these patients as systemic therapies are coupled with surgical management.”  Why limit and focus on urologists? Why not the multidisciplinary team caring for these patients?

Page 2, line 51; “…other immune checkpoint inhibitors have demonstrated encouraging results in the second-line treatment of mUC…” Suggest could introduce abbreviation here as opposed to later (line 58). ‘…other immune checkpoint inhibitors (CPI) have demonstrated encouraging results in the second-line treatment of mUC…’

Page 4, line 110; “Cell signaling is propegated via a tyrosine kinase…” Typo/spelling – suggest ‘Cell signaling is propagated via a tyrosine kinase…’

Author Response

Thank you for your comments. 

Overall Impression: The manuscript has been improved with revisions based on the previous reviewers’ comments and concerns. The presentation of the new agents alone or in combination, and their rationale for use, and ongoing clinical trial investigations is helpful to readers with an interest in urothelial cancer.

Major Concerns: It would be nice to know how new or novel drugs have been applied to other cancers and their response to help assess if there is a specific advantage to urothelial cancer with therapy being directed by next gen sequencing (NSG). This has been brought up before by the reviewers wanting to know if the drug was developed specifically for UC or another cancer. Based on the title “targeted therapies for urothelial cancers” this not beyond the scope of this paper as a significant proportion of drugs were developed for other indications.

Author response: Applications to other cancers have been added. Specifically: 1) Addition of a statement that FGFR mutations are common in both urothelial and cholangiocarcinoma 2) Erdafitinib for treatment of locally advanced, unresectable, or metastatic cholangiocarcinoma. 3) Infigratinib development originally for advanced cholangiocarcinoma and starting a trial for glioma 3) Expansion of phase 1 trial data on rogaratinib. 4) Comment of nectin-4 expression on epithelial cancers with mention that EV is only approved for mUC. 5) Addition of SG efficacy in triple negative breast cancer, NSCLC, and mUC with comment on FDA approval originally for BCa followed by accelerated approvals in NSCLC and mUC. 6) Discussed FDA approval of afatinib for NSCLC and some efficacy in HNSCC and metastatic BCa. 7) Added trastuzumab treatment for HER2+ breast ca, gastric ca, and GE junction ca. 8) Added HER2+ BCa for indications of tucatinib. 9) Commented on early promise of BA in multiple solid cancer types but the lack of efficacy leading to likely cessation of further investment by pharmaceutical companies 10) Added sentence on FDA approval for eganelisib in treatment of triple negative breast cancer 11) Added oncologic indications for everlimus 12) Added history of Buparlisib as a failed breast cancer agent 13. Added comment about Sapanisertib being under investigation in multiple cancer types with no current FDA approval

Minor Concerns: Being directed at urologists when the article should be directed at the multidisciplinary team caring for patients with mUC.

Author response: Mention of specific specialties removed

Line by line Review:

Page 1, line 25: “Urologists must be prepared to be involved in the multi-disciplinary care of these patients as systemic therapies are coupled with surgical management.”  Why limit and focus on urologists? Why not the multidisciplinary team caring for these patients?

Author response: This line has been removed as to not deter non-urologist readers.

Page 2, line 51; “…other immune checkpoint inhibitors have demonstrated encouraging results in the second-line treatment of mUC…” Suggest could introduce abbreviation here as opposed to later (line 58). ‘…other immune checkpoint inhibitors (CPI) have demonstrated encouraging results in the second-line treatment of mUC…’

Author response: The abbreviation has been moved up to line 51 as this is the first mention of checkpoint inhibitors.

Page 4, line 110; “Cell signaling is propegated via a tyrosine kinase…” Typo/spelling – suggest ‘Cell signaling is propagated via a tyrosine kinase…’

Author response: Thank you for the attention to detail. This has been fixed.

This manuscript is a resubmission of an earlier submission. The following is a list of the peer review reports and author responses from that submission.

Round 1

Reviewer 1 Report

Excellent, concise review and very well written, congratulations! Minor comments:

The Abstract is rather generic. Please amend a brief reference of the treatment principless that will be discussed, i.e. FGFR-inhibition, Nectin4-targeting etc.

Introduction P2 L39-40: Please quantify the percentage of cis-platin unfit patients. Citation?

P4 L150 Please add a word on the mechanism of action for trastuzumab plus pertuzumab, e.g. “HER2-targeted monoclonal antibodies trastuzumab plus pertuzumab“.

Antiangiogenic therapy? J Clin Oncol. 2018;36(15 suppl):4528.

Any other ADCs? J Clin Oncol. 2019;37(7 suppl):354.

P6 L220-221 Please add citations for single cell and spatial transcriptomics/proteomics and computational methods respectively

P6 L234 Please add citations for “new trials have begun advancing these targeted therapies earlier in the treatment paradigm”. Any of these new trials mentioned above?

Author Response

Thank you to the reviewer for the excellent comments.

The Abstract is rather generic. Please amend a brief reference of the treatment principless that will be discussed, i.e. FGFR-inhibition, Nectin4-targeting etc.  - Done

Introduction P2 L39-40: Please quantify the percentage of cis-platin unfit patients. Citation?  - Done (30-50%), added citation 

P4 L150 Please add a word on the mechanism of action for trastuzumab plus pertuzumab, e.g. “HER2-targeted monoclonal antibodies trastuzumab plus pertuzumab“. - Done, both are anti-HER2 monoclonal antibodies with different but complementary binding locations.

Antiangiogenic therapy? J Clin Oncol. 2018;36(15 suppl):4528. - We wanted to focus on targeted therapies that have arisen from next generation sequencing, antiangiogenic therapy was beyond the scope of this review.

Any other ADCs? J Clin Oncol. 2019;37(7 suppl):354. - Sacituzumab govitecan is described in the Trop-2 targeting paragraph and we focused on the larger, more recent studies that arose from Dr. Tagawa's earlier studies.

P6 L220-221 Please add citations for single cell and spatial transcriptomics/proteomics and computational methods respectively. - This is new and ongoing research with a vast array of different methodologies whose details are beyond the scope of this review but worth noting.

P6 L234 Please add citations for “new trials have begun advancing these targeted therapies earlier in the treatment paradigm”. Any of these new trials mentioned above? - These trials are primarily in development (including at our institution) and do not have citable publications yet.

Reviewer 2 Report

Targeted therapies in urothelial carcinoma – Review Article

Peter A. Reisz etal from Memorial Sloan Kettering Urological Services.

Journal MDPI Cancers -Open Journal since 2009  IF 6.129 (2019)

Line 31- 1. Introduction – ‘ numbering not needed as the rest of the headings are not numbered

The introduction is concise and well written establishing the natural history, nuances, and advances in research that could lead to gains in survival from metastatic UC (mUC).

Line 67- Next-generation sequencing identifies therapeutic targets-

This segment is also well written and concise however it doesn’t address basal or luminal phenotype and transition. Key markers of basal and luminal phenotype protein expression have been identified and used to guide therapy in clinical trials. This classification should be stated and not only referenced as a consensus document in molecular classification (ref. 17) because it is essential to the overall direction of this review paper. Would benefit on further discussion of the cell signaling pathways and their redundancy as well as commonality between cancers. Possibly this would best be presented in a cell signaling diagram /figure showing the pathways where the drugs they are discussing act and their potential overlap.

Line 87 - Fibroblast growth factor receptor (FGFR) inhibition

The authors introduce for the first time the luminal papillary phenotype molecular classification- but its significance is not known to the general audience of Oncology. Please see comment above that would facilitate general readership for a non-discipline specific oncology journal. To help the general readership understand the function of the TKIs downstream signaling and targets of the TKIs should be briefly presented (i.e. the MAP-Kinase pathway, STAT, PLC gamma, and the PI3K-AKT pathway). The downstream pathways are also helpful for the general reader to understand commonality of some of these agents – especially evident when the authors discuss HER2. The authors should comment that the side effects of the TKIs they are discussing are common to other TKIs.

Line 110 - Nectin-4 targeting

Again, the authors introduce for the first the molecular luminal subtype relate to onco-antigen Nectin-4. Please see comment above.  Pembrolizumab is introduced as an agent without a brief description of action for the general audience (i.e. ‘ pembrolizumab a PD1R blocker’). For the EV-303 study I would personally like to see the NCT # referenced (NCT03924895) consistently throughout the article.

Line 130 - Trop-2 targeting

This section is also well written and concise. Again, it would be my preference for the TROPiCS-04 Clinical Trial # NCT04527991 to be included – i.e. phase 3 clinical trial TROPiCS-04 (NCT04527991)- especially when there are multiple trials.

Line 142 - HER2 targeting

This section is also well written and concise. The HER2 down stream signalling is described but the authors reference a small phase 2 study (without NCT#) and don’t discuss where the drug has been used for other cancers especially breast, lung, and stomach cancers. I would again like both generic and trade names presented for better recognition by the general reader – i.e. afatinib (Gilotrif, Giotrif, Afanix) and trastuzumab (Herceptin) etc.. Again, deruxtecan is not presented in a way the general reader-ship would understand without referencing/searching all the time- suggest ‘deruxtecan – a topoisomerase inhibitor…’ this sets the mechanism action and the potential expected side effects in the readers mind. Again, there is inconsistency in reporting drugs and Trials that could easy improve the article. Another example is tucatinib suggest ‘…tucatinib – a HER2 protein TKI…’ this way the general audience understands the targets.

Line 160- TGF-b inhibition

Again, the section is concise and well written. However, it would be improved by adding in the NCT# for the referenced trials in this section.  

Line 175- PI3K inhibition

This section is too brief in its discussion of the PI3K/ AKT pathway and presents it in too simplex fashion. The AKT pathway is the survival pathway of almost all eukaryote cells and ultimately regulates apototic cell death through BCL2. This pathway does not need to be mutated to be constituently active in cancer. PI3K pathway is also involved in angiogenesis and cell migration. There is also a belief that it in regulates the immune system – bringing again to the forefront the redundancy of the cell signaling and targeted therapies. The mechanism of action of Eganelisib is not well presented so that the general reader would understand the authors reported combination.

Line 184 - mTOR inhibition

This section is again well written and concise. The conciseness misses that the TSC1/TSC2 is a heterodimer GPTase activating protein (GAP) that activates mammalian target of rapamycin complex 1 (TORC1) and mammalian target of rapamycin complex 2 (TORC2) [mTOR and 2 different adaptor proteins]. I would suggest changing “…regulation as well as angiogenesis via hypoxia-inducible factor-1 (HIF-1) and act via the tuberous sclerosis complex 1 and 2 (TSC1/TSC2).” to ‘…regulation as well as angiogenesis via hypoxia-inducible factor-1 (HIF-1) and act via the tuberous sclerosis complex 1 and 2 (TSC1/TSC2) GPTase activating protein (GAP).’ This change would make it clear that the subsequently discussed drugs are acting down stream.  The authors point out that 2 combination therapies with TKIs, CPI, and mTOR inhibitors have been initiated and are promising (also NCT# for reference). It would also be of benefit to the reader that dosing of TKIs to toxicity has been important in TKI studies.

Line 225 – Conclusions

Agree with the concluding remarks. However, there is no discussion on cisplatin cytotoxic chemotherapy and its effects on the immune system. This chemotherapeutic could upregulate cell death receptors or/and deplete and/or alter immune cells making for a more tolerogenic immune system. Artificial intelligence is unlikely to have a role unless the whole organism and cell populations are taken into consideration.

The authors concluding remarks “Further, this nuanced approach to therapy and patient selection underscores the importance of multi-disciplinary care of these patients, involving medical oncology, radiation oncology, and urology early in the disease course to identify the appropriate combination of systemic and surgical therapies for each patient” is not understood in the context of the systemic therapy presented. Radiation oncology and surgical oncology are targeted therapies. The abscopal effects of radiotherapy are poorly understood and were not discussed in the review. Tumor burden, and oligometastatic disease was not discussed in the context of mUC. The real teams presented here in my opinion are the medical oncologist and GU pathologist in a large center multidisciplinary cancer center participating in clinical trials. Center of excellence and improvement in GU oncology isn’t even discussed in the review. The team approach implied within this article with radiation oncology and urology would be early referral for mUC and planning less-toxic therapies for neoadjuvant and consolidative therapies. However, this was not discussed in the article nor was it really implied.

Overall, reasonably well written paper but seems to have been written for a word restricted journal. To the best of my knowledge MDPI does not have a word limit based on instruction to authors – “There is no fixed limit, but for papers longer than 10,000 words, please contact us before submitting your paper. Please note that we only edit research articles, not books or theses.” As such this review article could be fleshed out a bit to be of interest to the general audience rather than a focused GU speciality journal. The inconsistency in presenting clinical trials and drugs could be improved upon as detailed above.